# Immunohistochemical Changes in the Testicular Excurrent Duct System of Healthy, Male Japanese Quail (*Coturnix coturnix japonica*) Observed at 4, 6–7, 12, and 52 Weeks of Age

**DOI:** 10.3390/ijms232214028

**Published:** 2022-11-14

**Authors:** Mohammed I. A. Ibrahim, June H. Williams, Christo J. Botha

**Affiliations:** 1Department of Paraclinical Sciences, Faculty of Veterinary Science, University of Pretoria, Onderstepoort, Pretoria 0110, South Africa; 2Department of Veterinary Anatomy, University of West Kordofan, West Kordofan State, Gebaish P.O. Box 12942, Sudan

**Keywords:** age groups, cytoskeletal proteins, fibronectin, healthy Japanese quail, immunohistochemistry, testicular excurrent duct system, TUNEL assay

## Abstract

The immunolocalization of the cytoskeletal and the extracellular matrix proteins was investigated in the testicular excurrent duct system of healthy Japanese quail at 4, 6–7, 12 and 52 weeks of age. TdT dUTP Nick End Labeling (TUNEL) assay was used to assess apoptotic cell formation. The epithelia of the testicular excurrent duct system in birds of all age groups displayed various immunolabeling intensities and localization of cytokeratin 5 and beta-tubulin, while α-SMA was observed in epithelia only of 4-week-old birds. In all age groups, vimentin immunostaining was observed in the rete testes and efferent ductular epithelia, but not in the epididymal duct unit. The periductal smooth muscle cells of the excurrent duct system displayed variably intense immunopositivity with cytokeratin 5, desmin, fibronectin, α-SMA, and beta-tubulin. Furthermore, beta-tubulin and vimentin immunolabeled endothelial cells and fibroblasts with various intensities, while fibronectin immunostained extracellular matrices surrounding these cells. TUNEL-positive apoptotic cells were observed in the rete testes and efferent ductular epithelia, with increased frequency (*p* < 0.001) in 52-week-old birds. The study serves as a baseline normal for this region in healthy birds at 4, 6–7, 12, and 52 weeks of age, for comparison in future similar immunohistochemical studies involving environmental toxins affecting this region.

## 1. Introduction

The avian testicular excurrent duct system includes rete testis, proximal and distal segments of the efferent ducts, and the epididymal duct unit (i.e., the connecting duct, epididymal duct and ductus deferens) [1]. Spermatozoa are transported via these ducts as they mature and gain viability [2]. Cytoskeletal proteins that are present in the cells and peri- and interductal tissues of these regions [3,4,5,6] facilitate these functions. These proteins include, among others, cytokeratin 5, desmin, smooth muscle actin (SMA), vimentin, and tubulin [4,6,7].

Cytokeratin 5, a stratified epithelial keratin type, is expressed in the basal cells of squamous and glandular epithelia [8]. It has also been reported in the non-stratified epithelia of the efferent ducts, epididymis, and the ductus deferens of humans [9] and in the epididymis of the vampire bat [10], as well as in epithelial and myoepithelial cells of normal breast tissue of humans [11]. Cytokeratin 5 modulates the proliferation and differentiation of the cells in stratified epithelia [12].

Desmin intermediate filament is expressed in the cytoplasm of all muscle types and endothelial cells [13], while smooth muscle actin (SMA) is a contractile actin isoform present in smooth muscle cells [14] and myofibroblasts [15]. Immunohistochemical studies have demonstrated that desmin and SMA are localized in the peritubular smooth muscle layer of the epididymal duct of the Sunda porcupine [16], as well as in the testicular excurrent duct system of adult duck, emu, fowl, Japanese quail, masked weaver, ostrich, and turkey [3,4,6]. Desmin and SMA play an important role in the structural and functional support of the contractile action of smooth muscle cells [13,15].

The intermediate filament vimentin is present in the cytoplasm of cells of mesenchymal origin [17,18]. Vimentin has, however, been also recorded in cells of non-mesenchymal origin, such as the epithelial cells of the rete testis of the lesser mouse-deer, llama, and vicuna [19,20], and the rete testis and caput epididymis of the dog [21]. In avian species, vimentin is immunoexpressed in the epithelia of the testicular excurrent duct system of adult emu, fowl, Japanese quail, and turkey [3,6]. Vimentin is essential for vital mechanical and biological functions, including cell attachment, migration, signaling, and proliferation [18,22].

The microtubule tubulin has been demonstrated in spermatids and Sertoli cells in the testis of the rat [23], lesser mouse-deer [19], and bovine [24]. To our knowledge, no previous report has addressed the expression of tubulin in birds. Tubulin is involved in many essential processes in the eukaryotic cell, including signaling, reproduction and division, and intracellular transport, as well as movement, development, and maintenance of cell shape [25,26].

The epithelium of the ducts forming the testicular excurrent system is surrounded by the periductal smooth muscle layer and interductal tissue, which contains blood vessels and fibroblasts [1]. The sub-epithelial reticular layer underlying the basal lamina continues into the extracellular matrix surrounding and connecting smooth muscle- and endothelial cells and comprises collagen type IV, entactin, fibronectin, heparan sulphate proteoglycan, and laminin [27]. In addition to providing structural support, extracellular matrices play an essential role in the regulation of cellular differentiation, migration, survival, morphogenesis, and homeostasis [28,29].

The testicular excurrent duct system undergoes seasonal and age-related regression, which occurs via apoptosis [30,31]. Apoptosis is a natural physiological mechanism of programmed cell death that differs morphologically and biochemically from necrosis [32,33,34]. Cell death via apoptosis plays an essential role in the regulation of physiological function and homeostasis [32].

Immunohistochemistry has demonstrated that certain cytoskeletal proteins and laminin, an extracellular protein, are present in epithelial and peri- and interductal tissues of the excurrent duct system of various adult avian species [3,4,5,35]. There are no reported data on the expression and localization of the various cytoskeletal proteins and the extracellular protein fibronectin in the cellular components of these regions in birds of different age groups. The present study was, therefore, designed to provide baseline data on age-related changes in the immunolocalization of some cytoskeletal proteins and fibronectin in the normal testicular excurrent duct system of captive-bred and raised 4-week-old (immature), 6–7-week-old (post-pubertal), 12-week-old (adult) and 52-week-old (adult) Japanese quails. In addition, the apoptotic rates of cells in the normal rete testes and efferent ductular epithelia were determined.

The nature of the immunohistochemical labeling in the quail tissues was further compared and verified by labeling a variety of control tissues beyond those performed in previous studies [3,4,5,35]. In the current study, Japanese quail testicular excurrent duct system labeling, at each reproductive stage, was compared with the labeling of adult domestic canine, domestic chicken, and Japanese quail positive and negative control tissue samples for each of the antibodies studied, using the manufacturer’s instructions. The only antibodies for which manufacturers have verified labeling in birds, other than domestic species, were smooth muscle actin (avian) and vimentin (chicken) (Table 1).

## 2. Results

### 2.1. Immunohistochemistry

The detailed intensities of the immunolabeling and distribution of cytokeratin 5, desmin, fibronectin, α-SMA, beta-tubulin, and vimentin in various cells of all ducts forming the testicular excurrent duct system, including the rete testis, proximal efferent duct, distal efferent duct, connecting duct, epididymal duct, and ductus deferens, are summarized in Table 2 and Table 3.

#### 2.1.1. Cytokeratin 5


*
Epithelium
*


In all segments of the testicular excurrent duct system, cytokeratin 5 immunolabeled the epithelia with various intensities and localization (Figure 1a–g and Figure 2a–g). Spermatozoa were immunonegative for cytokeratin 5 (Figure 2d–g). In the positive controls, the epidermal layers in the skin of an adult Japanese quail and dog, as well as the epididymal region epithelium of the chicken, were cytokeratin 5 immunopositive (Figure 1h,i and Figure 2h,i).


*
Peri- and interductal tissues
*


There was variable cytokeratin 5 immunostaining of periductal smooth muscle layers of the testicular excurrent duct system in all birds of different age groups, with weak to moderate immunoreactivity observed in the interductal vascular endothelial cells, and fibroblasts were immunonegative (Figure 1b,c,e–g and Figure 2a–g). Cytokeratin 5 labeled the vascular endothelial cells of the epididymal region of chicken and the dermis layer of the dog (Figure 1i and Figure 2h), which were positive controls.

#### 2.1.2. Desmin


*
Epithelium
*


The epithelia and the luminal spermatozoa of the testicular excurrent duct system, in all birds of different age groups, were desmin immunonegative (Figure 3a–f).


*
Peri- and interductal tissues
*


In all birds of different age groups, there were variable intensities of desmin immunostaining observed in the periductal smooth muscle layers and tunicae media of interductal blood vessels of the testicular excurrent duct system, while the interductal vascular endothelial cells and fibroblasts in this region were immunonegative for desmin (Figure 3a–f). Immunopositive labeling in the smooth muscle layers of the small intestine of the adult Japanese quail and epididymal region of chicken, as well as in the straited muscle of the dog tongue (Figure 3g–i), supported the desmin-positive labeling detected in the peri- and interductal tissues of the excurrent duct system of the Japanese quails at different age groups.

#### 2.1.3. Fibronectin


*
Epithelium
*


There was negative fibronectin immunostaining in the epithelia and their underlying basement membranes in the ducts forming the testicular excurrent duct system of all birds in the different age groups (Figure 4a–g).


*
Peri- and interductal tissues
*


Fibronectin immunostaining was observed in the extracellular matrices associated with smooth muscle cells and fibroblasts forming the periductal smooth muscle layer, as well as vascular endothelial cells in the excurrent duct system of all birds of different age groups (Figure 4a–g). In the positive controls sections, there was positive fibronectin immunolabeling in the extracellular matrices of Bowman’s capsules and tubules in the kidneys of an adult Japanese quail and a dog, as well as in the peri- and interductal tissues of the chicken epididymal duct (Figure 4h,i).

#### 2.1.4. Alpha-Smooth Muscle Actin (α-SMA)


*
Epithelium
*


In 4-week-old birds, there was weak immunostaining of α-SMA in the epithelia of all ducts of the testicular excurrent duct system (Figure 5a,b). In other age groups, however, there was no epithelial α-SMA labeling of this system (Figure 5c–g). In addition, the luminal spermatozoa were immunonegative.


*
Peri- and interductal tissues
*


In all birds of different age groups, α-SMA was immunolocalized in the periductal smooth muscle cells layer and tunicae media of interductal blood vessels of the testicular excurrent duct system, but fibroblasts were α-SMA immunonegative (Figure 5a–g). This was supported by immunopositive α-SMA labeling observed in the smooth muscle layers of the small intestine of adult Japanese quail, epididymal duct of the chicken and esophagus of the dog (Figure 5h–j) used as positive control tissues.

#### 2.1.5. Beta-Tubulin


*
Epithelium
*


There were various intensities and different immunolocalizations of beta-tubulin in the epithelia of all ducts of the testicular excurrent duct system in the different age groups (Figure 6a–g and Figure 7a–e). The luminal spermatozoa were beta-tubulin immunopositive (Figure 7c–e). In addition, beta-tubulin was also detected by immunohistochemistry (Figure 6i and Figure 7f) and by immunofluorescence (Figure 8a–c) in the cilia of the efferent ductular epithelium and in spermatozoa in the testes of an adult Japanese quail and a chicken.


*
Peri- and interductal tissues
*


Beta-tubulin immunostaining was observed in the periductal smooth muscle layer and the endothelial cells, and tunicae media of the interductal blood vessels, as well as fibroblasts in the testicular excurrent duct system of all birds of different age groups (Figure 6a–g and Figure 7a–e).

#### 2.1.6. Vimentin


*
Epithelium
*


In all birds of different age groups, there were various intensities and immunolocalization of vimentin in the epithelia of the rete testis and proximal and distal segments of efferent ducts (Figure 9a–g). There was, however, no vimentin detected in the epithelium of the epididymal duct unit and the luminal spermatozoa (Figure 10a–f). Vimentin immunostaining was also detected in Sertoli cells and epididymis epithelium of dog and lymphocytes in the submucosal layer of the adult Japanese quail colon, as well as in the rete testis epithelium and a few epithelial cells in efferent ducts of chicken (Figure 9h,i and Figure 10h,i) used as positive control tissues.


*
Peri- and interductal tissues
*


In all birds of different age groups, the periductal smooth muscle layers of the excurrent duct system were immunonegative (Figure 9b,d,e,g and Figure 10a–f). There was, however, variably intense vimentin positivity observed in the vascular endothelial cells and fibroblasts in this region (Figure 9b,d,e,g and Figure 10a–f).

### 2.2. TUNEL-Stained Sections

In all age groups, apoptotic nuclei exhibiting varying intensities of TUNEL-staining occurred throughout the luminal epithelia of the rete testis and proximal and distal segments of the efferent ducts (Figure 11a–f). The mean and standard error of TUNEL positive nuclei in birds at 4, 6–7, 12, and 12 weeks of age were 1.30 (±0.20); 0.90 (±0.21); 1.33 (±0.22) and 6.33 (±0.61), respectively. The number of TUNEL-stained nuclei in the epithelial cells of the rete testis and efferent ducts in 52-week-old birds was significantly higher (*p* < 0.001) than in the other age groups (Figure 12).

## 3. Discussion

The present study represents the first report describing immunolabeling and localization of the cytoskeletal proteins cytokeratin 5, desmin, α-SMA, beta-tubulin, and vimentin, as well as fibronectin, an extracellular glycoprotein, in the normal cells and tissues of the testicular excurrent duct system of healthy, male Japanese quails in different age groups, from 4 weeks old to 52 weeks old. Positive immunolabeling was confirmed in control tissue samples collected from another adult, male Japanese quail, a chicken, and a dog. This supports the valid detection and interpretation of these cytoskeletal proteins and fibronectin in the excurrent duct system of Japanese quail. There was a variation in intensities of immunolabeling and the distribution of cytoskeletal proteins and fibronectin in epithelial cells and tissues of the testicular excurrent duct system in healthy Japanese quails at 4, 6–7, 12 and 52 weeks of age, suggesting that these proteins might be involved in the development and regulation of specific functions during sexual maturation. Other testing modalities, such as immunoblots, would be required to ratify the presence of these proteins.

The detection of apoptotic epithelial cells in the rete testis and efferent ducts by TUNEL labeling indicated a significant degree of apoptosis in this region in 52-week-old birds, which might also contribute to the decrease in the immunolocalization and intensity of labeling of the cytoskeletal proteins and fibronectin, which were observed in 52-week-old birds in this study. Choubey et al. [36] reported a decrease in adiponectin and its receptors’ expression and testosterone synthesis in the testis of aged mice, which leads to an increase in oxidative stress and alters testicular functions [37] Apoptosis can be induced by oxidative stress [38] and the morphology and functions of the excurrent duct system is regulated by steroid hormone activity [39]. Therefore, the increase in the apoptotic cell counts and the decrease in the immunolabeling of cytoskeletal proteins and fibronectin observed in this study might have been associated with adiponectin-altered testicular functions and increased oxidative stress.

### 3.1. Immunohistochemistry

#### 3.1.1. Cytokeratin 5

Cytokeratins, including cytokeratin 5, are the largest group of epithelium-specific intermediate cytoskeletal filaments [9,40] and fulfil an important role in the development and differentiation of epithelial cells [41]. Cytokeratins are also responsible for the protection of cells from mechanical and non-mechanical stresses [42]. In the current study, there was moderate to strong cytokeratin 5 immunostaining observed in the epithelia of all ducts forming the testicular excurrent duct system of 4-week-old, 6–7-week-old, and 12-week-old Japanese quail, but it was weakly to moderately immunopositive in similar regions of the 52-week-old birds (Figure 1 and Figure 2). In addition, cytokeratin 5 was also observed in the rete testis and epididymal duct epithelia of the chicken used as positive control (Figure 1h and Figure 2h). No previous reports are available concerning the presence specifically of cytokeratin 5 in the testicular excurrent duct system of birds; nonetheless, a cytokeratin (type not specified) was demonstrated in the epithelia of this region in adult fowl and turkey, while reportedly being immunonegative in the epithelium of the testicular excurrent duct system of the adult duck and Japanese quail [6]. 

Cytokeratin 5 is a Type II keratin that is produced by the basal cells in stratified epithelia [43], although it has been detected in non-stratified epithelia, including those of the efferent ducts, epididymal duct, and the ductus deferens of humans [9] and the epididymis of the vampire bat [10]. Bragulla and Homberger [43] speculated that the expression of cytokeratin 5 in the non-stratified epithelia suggests that it might be involved in the transportation of cytoplasmic membrane-bound vesicles. Thus, it is surmised that cytokeratin 5 observed in the testicular excurrent duct system epithelium in this study might facilitate transport functions in these cells. Cytokeratin 5 in these epithelia decreased in 52-week-old birds, while TUNEL assay revealed more apoptotic cells at 52 weeks of age (Figure 12).

Moderate cytokeratin 5 immunostaining observed in the interductal vascular endothelial cells of the testicular excurrent duct system from 4-week-old to 6–7-week-old to 12-week-old birds, with weak immunostaining in 52-week-old birds and similar presence in the endothelial cells of the positive control tissues collected from the epididymal region of a chicken (Figure 2h) and dog skin (Figure 1i), suggest the possibility of plasticity of these cells [44]. Further studies are warranted to investigate the role of the cytokeratin 5 intermediate filaments in the vascular endothelial cells in the testicular excurrent duct system of avian species.

#### 3.1.2. Desmin and α-SMA

In the current study, the periductal smooth muscle layers in the excurrent duct system were desmin and α-SMA immunopositive in all birds of the different age groups (Figure 3 and Figure 5 and Table 3). Similar observations have been reported in adult duck, fowl, Japanese quail, masked weaver, and turkey [4,6,7]. Positive control immunolocalization of desmin and α-SMA was confirmed in the current study in the small intestine of an adult Japanese quail, the epididymal duct of a chicken, and striated muscle of a dog. Desmin intermediate filaments and SMA microfilaments are involved in the mechanical support and contractile action of myoid cells [13,15]. The testicular excurrent duct system of birds is surrounded by periductal smooth muscle cells, which ensure that spermatozoa are transported via the ducts of this region, due to their contractions [4].

Furthermore, the current study indicated that desmin and α-SMA were immunoexpressed in the tunicae media of interductal blood vessels in all ducts of the testicular excurrent duct system. This was previously confirmed in an adult ostrich [5], duck, fowl, Japanese quail, and turkey [6]. Desmin and SMA are involved in regulating the contractile and dilatory functions of blood vessels [45,46].

Surprisingly, weak immunoexpression of α-SMA was demonstrated in the epithelia of all ducts forming the testicular excurrent duct system of birds at 4 weeks of age. Smooth muscle actin was also reported in the epithelia of the testicular excurrent duct system of the adult duck, fowl, and Japanese quail [6], as well as in efferent ducts and the epididymal duct unit of an adult emu [3]. It is well known that SMA is a myofibroblast marker and does not express in normal epithelial cells; however, SMA has also been reported in normal lens epithelial cells of the bovine, human, mouse, and rabbit [47], as well as in Sertoli–Sertoli junctions of ground squirrel testis [48]. Vogl and Soucy [48] concluded that the SMA in Sertoli cells is more structural than contractile. The α-SMA observed in the epithelial cells of the testicular excurrent duct system of Japanese quail at 4 weeks of age, therefore, could be functioning as structural cytoskeletal protein or be present as myoepithelial cells in the epithelia. Myoepithelial cells are modified epithelial cells that have characteristics similar to those of smooth muscle cells and provide support for the secretory cells [49]. Further studies are needed to investigate this phenomenon.

#### 3.1.3. Fibronectin

Fibronectin is a large molecular weight glycoprotein that represents the major extracellular matrix component [50,51] and is involved in embryogenesis and cell adhesion, migration, and growth [52,53]. In the current study, for all birds at different age groups, fibronectin was not observed in the epithelial basement membrane extracellular matrices of the testicular excurrent duct system. In the positive controls, although similar regions of Bowman’s capsule in the kidney of adult Japanese quail and dog were fibronectin immunopositive (Figure 4i,j), this was not detected in the epididymal region of the chicken (Figure 4h). In all birds of different age groups, however, fibronectin was immunolocalized in the extracellular matrices connected to smooth muscle cells in the periductal layer of the testicular excurrent duct system, with intensity decreasing with age (Figure 4). Fibronectin has been detected in the smooth muscle cells in the esophagus, stomach, jejunum, and urinary bladder of humans [54] and in the epididymis of adult Long–Evans rats [55]. In the current study, fibronectin was also detected in extracellular matrices associated with the interductal vascular endothelial cells of all birds of different age groups (Figure 4). Fibronectin can be produced by fibroblasts, myoblasts, astrocytes, and hepatocytes, as well as by kidney and endothelial cells [56]. It was difficult to see whether the immunostaining of fibronectin was in the cytoplasm of the endothelial cell, but further studies using higher magnification, such as immunoelectron microscopy, are needed to investigate the presence of fibronectin in this cell. In humans, fibronectin has been observed in the vascular endothelial cells in various tissues including the epididymis [51]. The fibronectin in the extracellular matrices associated with smooth muscle and vascular endothelial cells of the testicular excurrent duct system in this study could play a role in the regulation of these cellular functions [52,53].

#### 3.1.4. Beta-Tubulin

In the present study, there was strong beta-tubulin immunostaining in the epithelial cells of the rete testis of 4-week-old (Figure 6a), 6–7-week-old (Figure 6c), and 12-week-old birds. In 52-week-old birds, however, similar areas of epithelia in this region were only moderately beta-tubulin immunopositive (Figure 6f). It has been reported that acetylation of tubulin increased with age in the cerebellum of Wistar rats [57]. Acetylation is a process that induces a significant change in the protein, including degradation [58], and so the possibility exists that the decrease in tubulin staining observed in the rete testis epithelium of 52-week-old birds may be due to increased acetylation in this region. Tubulin immunostaining has been reported in the rete testis epithelium of the bovine from the post-natal stage (8 weeks old) to the adult stage (72 weeks old) [24], but was absent in the epithelium of the rete testis in the adult lesser mouse-deer [19].

In all birds of different age groups in the current study, beta-tubulin was demonstrated in the cytoplasm of the cells lining the proximal and distal segments of efferent ducts (Figure 6), as well as in the epididymal duct unit (Figure 7). In addition, beta-tubulin immunostaining was detected in the cilia of the ciliated cells lining the efferent ducts (Figure 6) and in luminal spermatozoa, which was confirmed by immunofluorescence (Figure 8). In adult Wistar rats, α-tubulin immunostaining in the cilia of ciliated cells in this region has been described [59], whereas the antibody against microtubule-associated protein 1B has been reported to immunostain the cilia and cytoplasm of ciliated cells in the efferent ducts [60]. Tubulin is important for the regulation, signaling, and motility of cilia [61]. The efferent ductular motile cilia maintain the immotile spermatozoa, suspended in the luminal fluid, through their continuous movement in different directions [62]. Thus, they play an important role in the transportation of spermatozoa from the rete testis to the epididymis, where spermatozoa gain viability and maturity [39].

In the current investigation, beta-tubulin was immunoexpressed in the tunicae media in interductal blood vessels of the testicular excurrent duct system. The moderate intensity of beta-tubulin immunoreaction during the immature stage (4 weeks old) decreased as the birds matured sexually (Table 3). Tubulin immunoreactivity in the tunica media has previously been demonstrated in the cerebral blood vessels of adult, male Sprague–Dawley rats and C57 Black 6 mice [63]. Pritchard et al. [63] suggested that tubulin plays an important role in regulating contractility of the cerebral blood vessels by maintaining the peripheral connection of smooth muscle cells.

#### 3.1.5. Vimentin

Vimentin is a major intermediate filament that is expressed in cells of mesenchymal origin. Furthermore, it is also present in cells of non-mesenchymal origin, including the Sertoli cells of humans and the Wistar rat [64,65] and the epididymis and ductus deferens epithelium in man [65]. In the current study, vimentin was observed in the Sertoli cells of the dog testis used as a control positive for vimentin immunohistochemistry (Figure 9h). Vimentin is involved in the positioning of the nucleus of the Sertoli cell [66] and plays an important role in the organization of cellular attachment, migration, and signaling [18], supporting elasticity and protection from mechanical stress in the epithelial cells [67].

Previous studies in adult fowl, Japanese quail, turkey [6], and emu [3] demonstrated vimentin in the epithelia of the rete testis, efferent ducts, and the epididymal duct unit. Conversely, vimentin immunostaining was absent in the testicular excurrent duct system epithelium of the adult duck and the masked weaver [4,6]. In the current study, in all birds at different age groups, there were various intensities of immunolabeling and immunolocalization of vimentin in the epithelia of the rete testis and efferent ducts, as well as in the chicken used as a positive control (Figure 9). It is concluded that immunoexpression of vimentin in the testicular excurrent duct system of birds may be species-specific.

Vimentin was not observed in the epididymal duct unit epithelia of Japanese quails at different age groups, which concurs with studies conducted in the adult duck, emu, ostrich, and masked weaver [4,68] (Figure 10). In contrast, Aire and Ozegbe [6] reported that vimentin was immunoexpressed in the epithelium of the epididymal duct unit of the adult fowl, Japanese quail, and turkey. This inconsistency could be ascribed to the immunostaining kit used in their study, but it also could be due to the fact that the cells lining the epididymal duct unit are not of mesenchymal origin.

Fibroblasts originate from mesenchymal cells and are widely distributed in various tissues and organs [69]. There are no previous reports on the presence of vimentin in the fibroblasts of peri- and interductal tissues in the avian testicular excurrent duct system. In the current study, moderate to strong vimentin immunostaining was observed in the fibroblasts of peri- and interductal tissues in the testicular excurrent duct system in 4-week-old, 6–7-week-old, and 12-week-old birds (Figure 10). In 52-week-old birds the fibroblasts were immunonegative for vimentin. The presence of vimentin intermediate filaments in fibroblasts has been described in the testis of post-hatch Japanese quail [70] and adult black-backed jackal [71].

### 3.2. TUNEL-Stained Sections

Apoptosis is programmed cell death and considered as a protective mechanism of body cells against the accumulation and distribution of defective cells [72,73]. It plays an essential role in protecting the body against age-associated tumorigenesis [72]. The current study demonstrated the presence of TUNEL-positive epithelial cells in the rete testis and efferent ducts of Japanese quail in the different age groups. TUNEL-positive cells were significantly more numerous (*p* < 0.001) in 52-week-old birds than in the other age groups. Higher frequencies of apoptotic cell counts have also been reported in the epididymis of 2-year-old mice, compared with 3-month-old mice [74]. Jara et al. [74] attributed the increase in apoptotic cells to the reduction in testosterone concentrations and suggested that testosterone diminishes the apoptotic processes. Plasma testosterone concentrations are lower in 73-week-old chickens [75]; thus, the higher apoptotic rate observed in the epithelia of the rete testis and the efferent ducts of 52-week-old birds in this study could be due to the lower testosterone levels. However, it would have been valuable to study oxidative stress markers and to correlate them with the presence of TUNEL-positive cells in these regions at different age groups.

## 4. Materials and Methods

### 4.1. Animals and Tissue Preparation

A total of 28 healthy, male Japanese quail (*Coturnix coturnix japonica*) consisting of four groups (*n* = 7)—4-week-old birds (immature), 6–7-week-old birds (post-pubertal), and 12-week-old and 52-week-old birds (adult groups)—were used in the study. The age group classifications were based on those reported by [76]. The birds were purchased from the Aviary Unit, Irene Animal Improvement Research Station, Pretoria, Gauteng Province, South Africa. After hatching, the birds were brooded and reared in battery cages (46 cm × 95 cm × 51 cm) under a controlled photoperiod cycle (16 h light: 8 h dark). The room temperature was maintained at 25 ± 2 °C. The birds were fed a standard commercial high-protein diet (Obaro Feeds, Pretoria, South Africa) with free access to fresh water. Puberty was determined as the first day of release of cloacal foam [77]. The birds were serially euthanized as they grew older, using carbon dioxide (CO_2_) inhalation. The epididymal regions (rete testis, both segments of efferent ducts, the connecting duct and the epididymal duct), and the ductus deferens were immediately removed from the testes (left and right) and fixed in 10% buffered formalin. Then, tissue samples were routinely processed for light microscopy. The study was approved by the University of Pretoria’s Animal Ethics Committee (approval number V034-18).

Subsequently, a sexually active, mature male Japanese quail was obtained from Zelda Enslin Farm, Pretoria, Gauteng Province, South Africa, and euthanized by cervical dislocation (Animal Ethics Committee approval number REC128-20). Various normal tissues were immediately collected and fixed in 10% buffered formalin. Tissue samples were routinely processed for light microscopy. Normal chicken and domestic canine tissue samples embedded in wax-blocks stored at the Section of Pathology, Immunohistochemistry Laboratory, were used as positive and negative controls for each antibody and for comparison with the labeling in the Japanese quail tissues.

### 4.2. Immunohistochemistry

The immunohistochemistry was performed on sections (3 µm thick) on superfrost glass slides using a Biogenex super-sensitive one-step polymer-HRP detection kit (Emergo Europe, Hague, Netherlands). The sections were deparaffinized and endogenous peroxidase activity was blocked using a 3% (*v*/*v*) H_2_O_2_ in phosphate-buffered saline solution (PBS, pH 7.6) for 5 min. For antigen retrieval, sections were microwaved at 750 W for three cycles of 7 min each in citrate buffer (pH 6) or Tris-EDTA (pH 9). The slides were then incubated with specific primary antibodies in a humidified chamber at 37 °C for 1 h. The antibodies used were cytokeratin 5, desmin, fibronectin, α-SMA, beta-tubulin, and vimentin (Table 1). The slides were rinsed twice in PBS for 5 min each and then incubated in a humidified chamber at 37 °C with a polymer-HRP reagent (Emergo Europe, Hague, Netherlands) for 15 min. The slides were subsequently washed thrice in PBS for 5 min each. Reactivity was visualized by applying a 3,3′-diaminobenzidine solution (Emergo Europe, Hague, Netherlands). Sections were counter-stained with Mayer’s hematoxylin for 1 min. The sections were dehydrated, cleared in xylene, and mounted with dibutyl phthalate polystyrene xylene (Sigma-Aldrich, Johannesburg, South Africa). Sections were examined using an Olympus BX-63 light microscope with an attached digital camera for image capture.

Negative control sections included in each set of figures were processed identically, except that the antibody was replaced with PBS. In the positive control samples, for each antibody used, various normal tissue sections from Japanese quail were compared to simultaneous immunolabeling in known-positive domestic canine and a healthy domestic chicken. Digital images of negative and positive staining are included in each set of figures.

Based on the visual examination, the relative intensities of immunolabeling of cytokeratin 5, desmin, fibronectin, α-SMA, beta-tubulin, and vimentin were identified as negative (−), weak (+), moderate (++), and strong (+++), as described by Madekurozwa and Kimaro [78], and tabulated (Table 2 and Table 3).

### 4.3. Immunofluorescence for Beta-Tubulin-Cy3 Antibody

The epididymal region and testis samples were obtained from Japanese quail and chicken and fixed immediately in 10% buffered formalin for 3 days. Tissue samples were routinely processed for light microscopy. The immunofluorescence was performed on sections (3 µm thick) on superfrost glass slides. The sections were deparaffinized and endogenous peroxidase activity was blocked using a 3% (*v*/*v*) H_2_O_2_ in phosphate-buffered saline solution (PBS, pH 7.6) for 5 min. For antigen retrieval, sections were microwaved at 750W for three cycles of 7 min each in Tris-EDTA (pH 9). Then, the slides were incubated with mouse monoclonal anti-beta tubulin-Cy3 antibody (Sigma C4585), at 37 °C for 1 h and at dilution 1:100 in PBS (Sigma P4417). The slides were rinsed thrice in PBS for 5 min each while agitating gently after, which sections were incubated with nuclear counterstain 4′,6-diamidino-2-phenylindole dihydrochloride (DAPI, Sigma D9542) (1.3 µg/mL) at 37 °C for 15 min. The slides were subsequently rinsed, as described previously, in PBS. The sections were mounted with Prolong Gold Antifade (Invitrogen P36930) and sealed with clear nail varnish to prevent the sample from drying out. Images were captured using an Olympus BX-63 fluorescent microscope with an attached digital camera.

### 4.4. TdT dUTP Nick End Labeling (TUNEL) Assay

Apoptotic cells were labeled on 3 µm thick sections using an ApopTag plus peroxidase in situ apoptosis detection kit, based on the manufacturer’s instructions (Millipore, Temecula, CA, USA). Rat mammary gland provided by Millipore, Temecula, CA, USA, was used as a positive control.

### 4.5. TUNEL-Positive Cell Counting

The number of TUNEL-positive nuclei in 100 μm lengths of the rete testis and efferent ducts from 10 randomly selected microscopic fields per section of birds (*n* = 3) per age group were counted. An image-analyzer system (CellSens dimension software, Tokyo, Japan) linked to an Olympus BX-63 microscope (Tokyo, Japan) was used to count the TUNEL-positive nuclei. This was conducted according to the method described by Mpango and Madekurozwa [79].

### 4.6. Statistical Analysis

The number of apoptotic nuclei in the epithelium of the rete testis and efferent ducts of the Japanese quails at different age groups (4 weeks of age, 6–7 weeks of age, 12 weeks of age, and 52 weeks of age) was analyzed using one-way analysis of variance (ANOVA) (SPSS 23 software). Statistical significance was set at *p* < 0.05.

## 5. Conclusions

The current study documented that the intensities of immunolabeling and distributions of cytokeratin 5, desmin, fibronectin, smooth muscle actin, tubulin, and vimentin in the cells and tissues of the testicular excurrent duct system of healthy, captive-bred and raised Japanese quails were generally age-dependent. The immunolocalization of the cytoskeletal proteins and fibronectin was confirmed by using tissue samples obtained from an adult Japanese quail, as well as from a chicken and dog. In addition, the current study indicated that the frequencies of TUNEL-positive cell counts in the epithelia of the rete testes and efferent ducts were higher (*p* < 0.001) in 52-week-old Japanese quails than in those of earlier ages, namely immature (4 weeks old), post-pubertal (6–7 weeks old), and sexually active adults (12 weeks old). The information obtained from this study will be useful for reference in future investigations of the effects of hormones and environmental toxins, especially endocrine disruptors, on cells and tissues of the testicular excurrent duct system of Japanese quail, which is an excellent environmental sentinel species and experimental model.

## Figures and Tables

**Figure 1 ijms-23-14028-f001:**
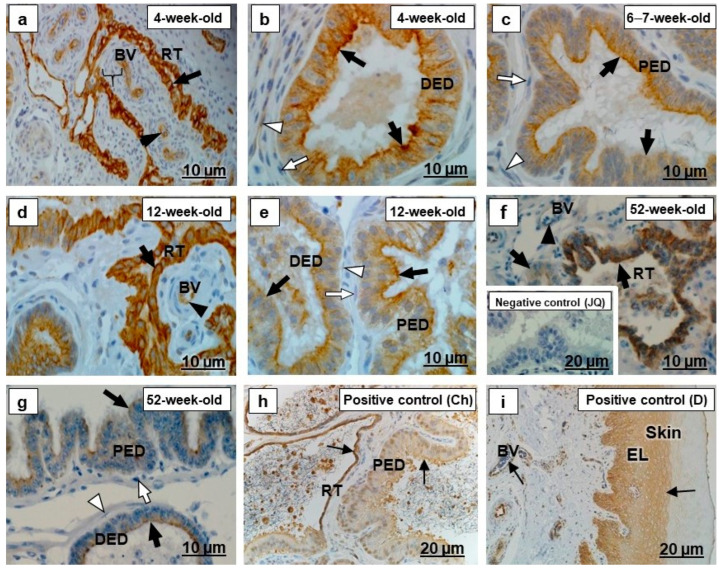
Immunolocalization of cytokeratin 5 in the (RT) rete testis, (PED) proximal efferent duct and (DED) distal efferent duct of the testicular excurrent duct system of Japanese quails (*n* = 7 per age group). (**a**,**d**) Strong immunostaining in the RT epithelia (black arrows). Moderate immunostaining in the vascular endothelial cells (black arrowheads). (**a**) Bracket shows weak immunostaining in the tunica media of the BV. (**b**) Strong immunostaining in the apical and subapical regions (black arrows) in the epithelia of the DED. Weak immunostaining in the smooth muscle cell (white arrowhead). (**c**,**e**) Moderate to strong immunostaining in the apical and subapical epithelial regions (black arrows) of the PED and DED. White arrowheads indicate negative to weak immunostaining of the smooth muscle cells. (**f**) Weak to moderate immunostaining in the RT epithelium (black arrows). Weak immunostaining in the vascular endothelial cell (black arrowhead). (**g**) Weak to moderate immunostaining in the apical epithelial regions (black arrows) of the PED and DED. Immunonegative smooth muscle layer (white arrowhead). (**b**,**c**,**e**,**g**) Immunonegative fibroblasts (white arrows). Inset (**f**) Negative control: Normal efferent ducts of an adult Japanese quail. (**h**,**i**) Positive controls: Thin black arrows indicate immunopositive labeling in the RT and PED epithelia of the adult chicken (**h**), as well as vascular endothelial cells and epidermal layer (EL) of adult dog skin tissue (**i**). BV, Blood vessel. Ch, Chicken. D, dog. JQ, Japanese quail.

**Figure 2 ijms-23-14028-f002:**
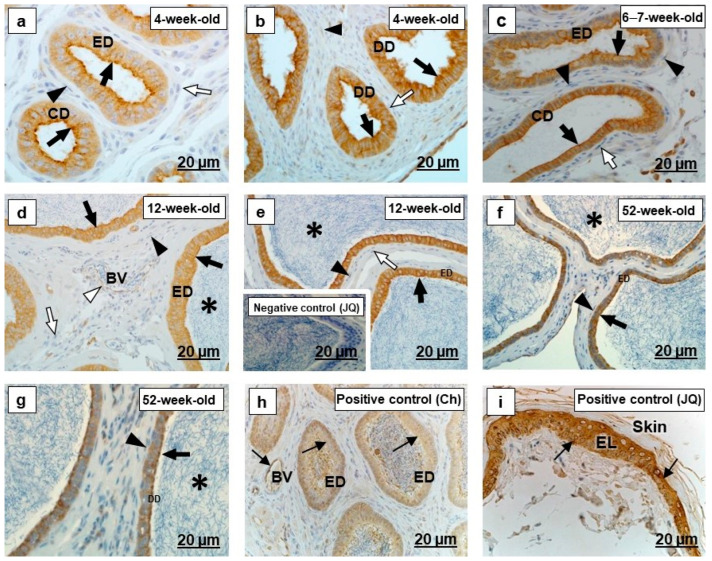
Immunolocalization of cytokeratin 5 in the epididymal duct unit- (CD) connecting duct, (ED) epididymal duct and (DD) ductus deferens of the testicular excurrent duct system of Japanese quails (*n* = 7 per age group). (**a**–**e**) Strong immunostaining in the epithelia (black arrows) of the CD, ED and DD. Immunonegative fibroblasts (white arrows). (**a**,**b**) Black arrowheads indicate weak immunostaining in the smooth muscle cells. (**f**,**g**) Moderate immunostaining in the epithelia (black arrows) of the ED and DD. (**c**–**g**) Negative to weak immunostaining in the smooth muscle cells (black arrowheads). (**d**) Moderate immunostaining in the vascular endothelial cell (white arrowhead). (**d**–**g**) Immunonegative luminal spermatozoa (asterisks). Inset (**e**) Negative control: Epididymal duct of an adult Japanese quail. (**h**,**i**) Positive controls: Thin black arrows indicate immunopositive labeling in the epididymal duct epithelium and vascular endothelial cells of the adult chicken (**h**) and epidermal layer (EL) of an adult Japanese quail skin tissue (**i**). BV, Blood vessel. Ch, Chicken. JQ, Japanese quail.

**Figure 3 ijms-23-14028-f003:**
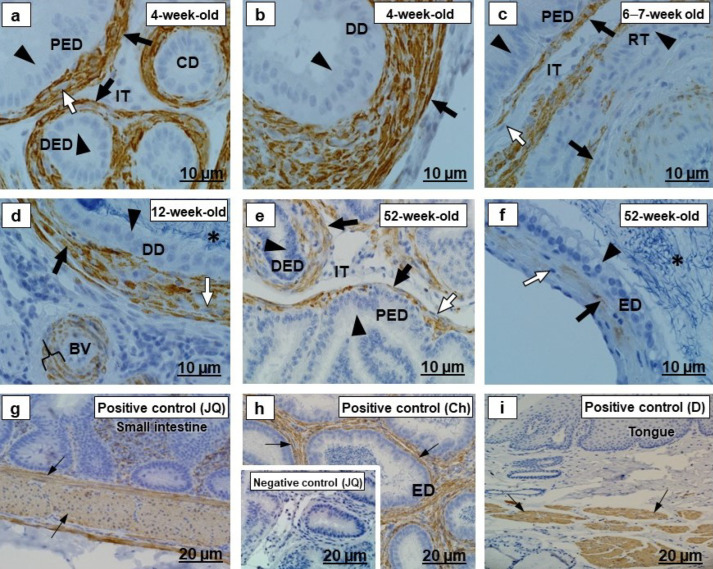
Desmin immunolocalization in the testicular excurrent duct system of Japanese quails (*n = 7* per age group). (**a**–**f**) Immunonegative epithelial cells (arrowheads), fibroblasts (white arrows) and the interductal tissue (IT) of the testicular excurrent duct system. (**a**,**b**) Black arrows indicate strong immunostaining in the periductal smooth muscle layers. (**c**,**d**) Moderate to strong immunostaining in the periductal smooth muscle layers (black arrows). (**d**) Bracket shows moderate immunostaining in the tunicae media of the interductal blood vessel. (**d**,**f**) Immunonegative spermatozoa (asterisks). (**e**,**f**) Black arrows depict weak to moderate immunostaining in the periductal smooth muscle layers. Inset (**h**) Negative control: Efferent ducts of an adult Japanese quail. (**g**–**i**) Positive controls: Thin black arrows depict immunopositive labeling in the smooth muscle layers of the small intestine of the adult Japanese quail (**g**), epididymal duct of the chicken (**h**), as well as in striated muscle in the tongue of the dog (**i**). RT, Rete testis. PED, Proximal efferent duct. DED, Distal efferent duct. CD, Connecting duct. ED, Epididymal duct. DD, Ductus deferens. BV, Blood vessel. Ch, Chicken. D, dog. JQ, Japanese quail.

**Figure 4 ijms-23-14028-f004:**
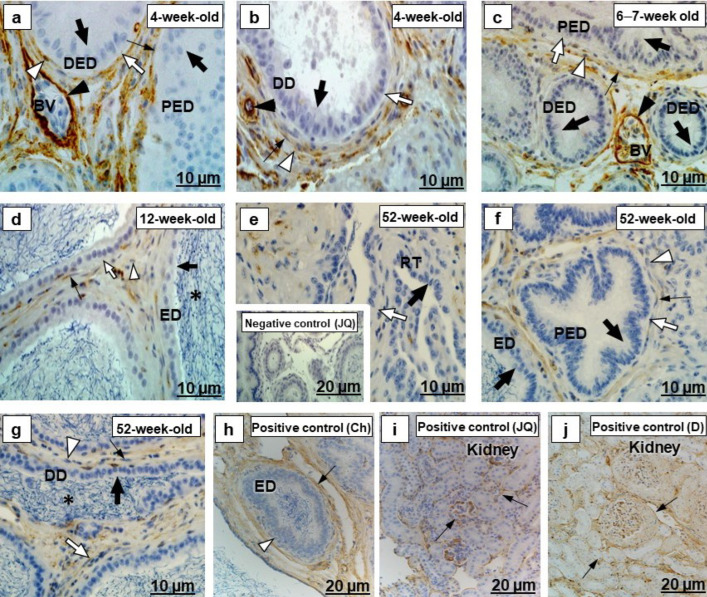
Immunolocalization of fibronectin in the testicular excurrent duct system of Japanese quails (*n = 7* per age group). (**a**–**g**) Immunonegative epithelia (thick black arrows) and their underlying basement membranes (thick white arrows) of the rete testis (RT), proximal efferent duct (PED), distal efferent duct (DED), connecting duct (CD), epididymal duct (ED) and ductus deferens (DD). (**a**–**d**) Moderate immunostaining in extracellular matrices associated with smooth muscle cells (thin black arrows) and fibroblasts (white arrowheads). (**a**–**c**) Black arrowheads depict strong immunostaining in extracellular matrices connected to endothelial cells of the BV. (**f**,**g**) Weak immunostaining in extracellular matrices connected to smooth muscle cells (thin black arrows) and fibroblasts (white arrowheads). (**d**,**g**) Immunonegative spermatozoa (asterisks). Inset (**e**) Negative control: Efferent ducts of an adult Japanese quail. (**h**,**i**) Positive controls: Thin black arrows indicate immunopositive labeling in peri- and interductal tissues of chicken epididymal duct (**h**), as well as in extracellular matrices of Bowman’s capsules and tubules in the kidney of the adult Japanese quail (**i**) and dog (**j**). (**h**) Immunonegative epithelia of the chicken epididymal duct (white arrowhead). BV, Blood vessel. Ch, Chicken. D, dog. JQ, Japanese quail.

**Figure 5 ijms-23-14028-f005:**
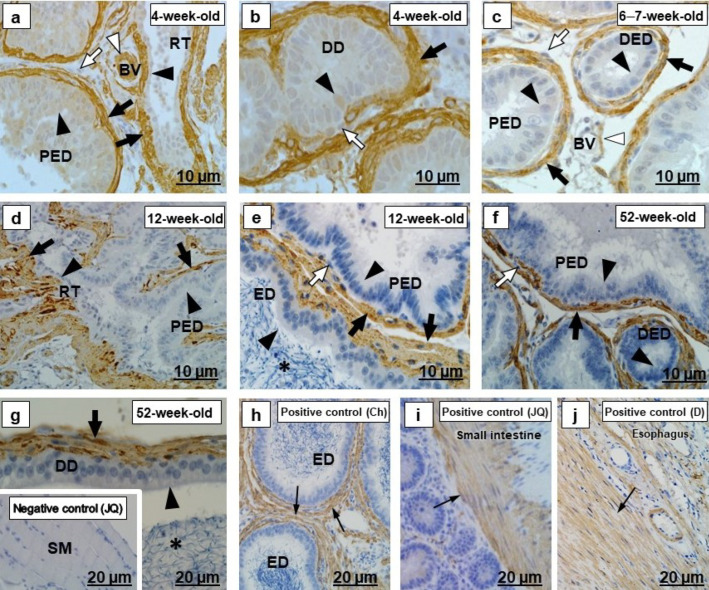
Alpha-smooth muscle actin immunolocalization in the testicular excurrent duct system of Japanese quails (*n* = 7 per age group). (**a**,**b**) Weak immunostaining in the epithelia (black arrowheads) of the RT, PED and DD. (**a**) Strong immunostaining in the tunica media (white arrowhead) of interductal blood vessels. (**c**–**g**) Immunonegative epithelia (black arrowheads) of the RT, PED, DED, ED and DD. (**a**–**g**) Black arrows indicate strong immunostaining in the periductal smooth muscle layer. Immunonegative fibroblasts (white arrows). (**c**) Moderate immunostaining in the tunica media (white arrowhead) of interductal blood vessel. (**e**,**g**) Immunonegative spermatozoa (asterisks). Inset (**g**) Negative control: Striated muscle of an adult Japanese quail. (**h**–**j**) Positive controls: Thin black arrows indicate immunopositive labeling of smooth muscle cells in the epididymal duct of the chicken (**h**), small intestine of an adult Japanese quail (**i**) and esophagus of the dog (**j**). RT, Rete testis. PED, Proximal efferent duct. DED, Distal efferent duct. CD, Connecting duct. ED, Epididymal duct. DD, Ductus deferens. BV, Blood vessel. Ch, Chicken. D, Dog. JQ, Japanese quail.

**Figure 6 ijms-23-14028-f006:**
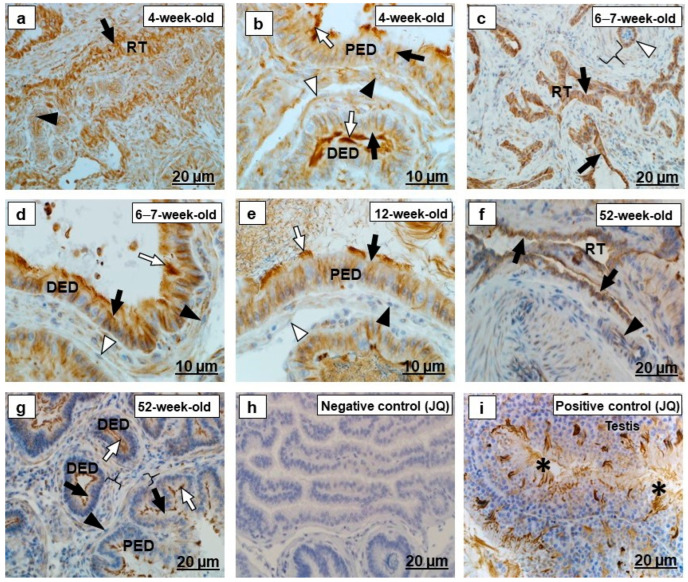
Beta-tubulin immunolocalization in the (RT) rete testis, (PED) proximal efferent duct and (DED) distal efferent duct of the testicular excurrent duct system of Japanese quails (*n* = 7 per age group). (**a**) Strong immunostaining in the RT epithelium (black arrow) and vascular endothelial cells (black arrowhead). (**b**) Moderate to strong immunostaining in the periductal smooth muscle cell (white arrowhead) and fibroblasts (black arrowheads). (**c**) Moderate to strong immunostaining in the epithelium (black arrows) of the RT. Moderate immunostaining in the vascular endothelial cells (white arrowhead). Weak immunostaining in the tunica media of interductal blood vessel (bracket). (**b**,**d**,**e**,**g**) White arrows depict strong immunostaining in the cilia. (**b**,**d**,**e**) Moderate to strong immunostaining in the lateral and subapical regions of cells (black arrows) lining the PED and DED. (**d**,**e**) Weak immunostaining in the periductal smooth muscle cell layers (white arrowheads) and fibroblasts (black arrowheads). (**f**) Weak to moderate immunostaining in the RT epithelium (black arrows) and the vascular endothelial cells, and tunica media (black arrowhead). (**g**) Weak immunostaining in the epithelium (black arrows), periductal smooth muscle layers (brackets) and fibroblasts (black arrowheads) of the PED and DED. (**h**) Negative control: Efferent ducts of an adult Japanese quail. (**i**) Positive control: Immunopositive sperm (asterisks) in the testis of an adult Japanese quail. JQ, Japanese quail.

**Figure 7 ijms-23-14028-f007:**
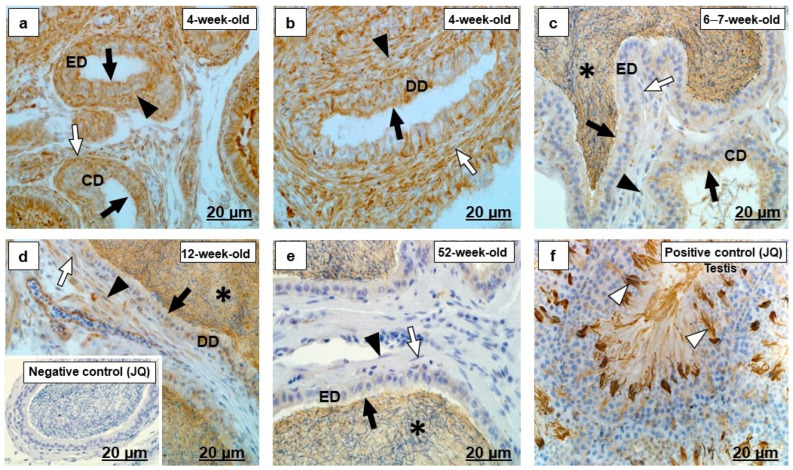
Immunolocalization of beta-tubulin in the epididymal duct unit- (CD) connecting duct, (ED) epididymal (DD) duct and ductus deferens of the testicular excurrent duct system of Japanese quails (*n* = 7 per age group). (**a**,**b**) Strong immunostaining in the epithelia (black arrows) of the CD, ED and DD. Moderate to strong immunostaining in the periductal smooth muscle cells (white arrows) and fibroblasts (black arrowheads). (**c**,**d**) Moderate to weak immunostaining in the epithelia (black arrows) of the CD, ED and DD. Moderate to weak immunostaining in the periductal smooth muscle cells (white arrows) and fibroblasts (black arrowheads). (**e**) Weak immunostaining in the ED epithelium (black arrow). Immunonegative periductal smooth muscle cell (white arrowhead) and fibroblast (black arrowhead). (**c**–**e**) Asterisks depict immunostaining of the luminal spermatozoa. Inset (**d**) Negative control: Epididymal duct of an adult Japanese quail. (**f**) Positive control: Immunopositive sperm (white arrowheads) in the testis of an adult Japanese quail. JQ, Japanese quail.

**Figure 8 ijms-23-14028-f008:**
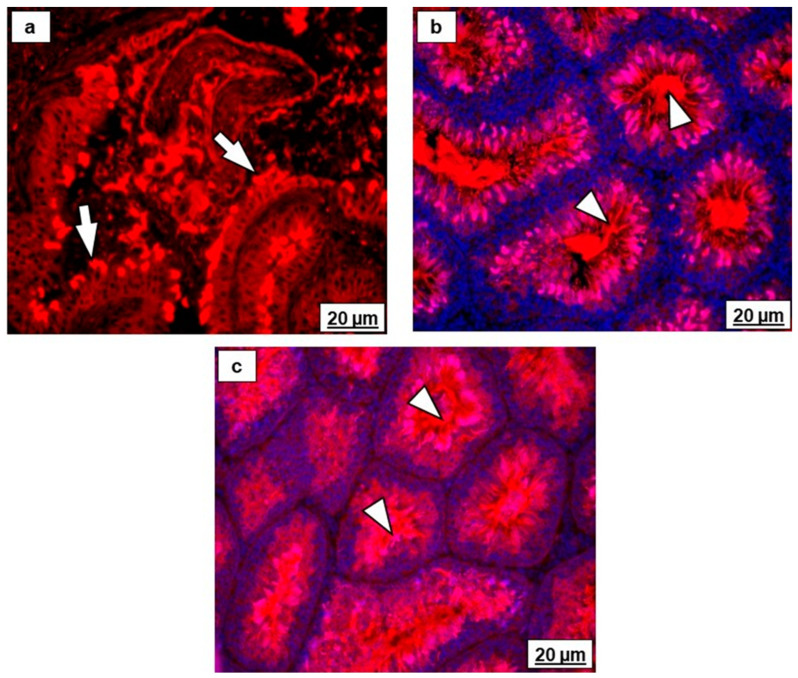
Immunofluorescence of beta-tubulin-Cy3 antibody in the (**a**) efferent ducts, (**b**) testis of an adult Japanese quail, and (**c**) chicken testis (*n* = 1 per animal). (**a**–**c**) Red signal- beta-tubulin immunostaining in the cilia of efferent ducts epithelium (white arrows) and spermatozoa of the testis (white arrowheads). (**b**,**c**) Immunostaining with 4′, 6-diamidino-2-phenylindole dihydrochloride (DAPI) in nuclei of the testicular epithelia (blue signal).

**Figure 9 ijms-23-14028-f009:**
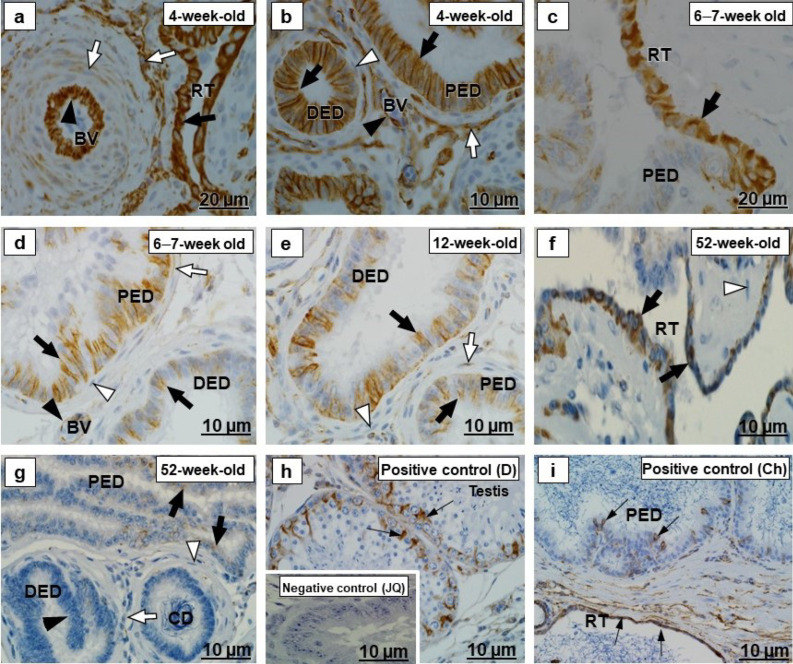
Immunolocalization of vimentin in the (RT) rete testis, (PED) proximal efferent duct and (DED) distal efferent duct of the testicular excurrent duct system of Japanese quails (*n* = 7 per age group). (**a**,**b**) Strong immunostaining in the epithelial (black arrows), and vascular endothelial cells (black arrowheads), as well as fibroblasts (white arrows) of the RT, PED, and DED. (**c**) Black arrow indicates moderate to strong immunostaining in the RT epithelium. (**d**,**e**) Moderate immunostaining in the ciliated cells (black arrows) of the PED and DED. White arrows depict moderate to weak immunostaining in the fibroblasts. (**f**) Weak to moderate immunostaining in the epithelium (black arrows) of the RT. (**g**) Black arrows show weak immunostaining in a few cells of the PED, immunonegative DED epithelium (black arrowhead) and fibroblast (white arrow). (**b**,**d**–**g**) Immunonegative smooth muscle cells (white arrowheads). Inset (**b**) Negative control: Efferent ducts of an adult Japanese quail. (**h**,**i**) Positive controls: Thin black arrows indicate immunopositive labeling in the Sertoli cells of the dog testis (**h**) and epithelial cells of the RT and PED of the chicken (**i**). CD, Connecting duct. BV, Blood vessel. Ch, Chicken. D, Dog. JQ, Japanese quail.

**Figure 10 ijms-23-14028-f010:**
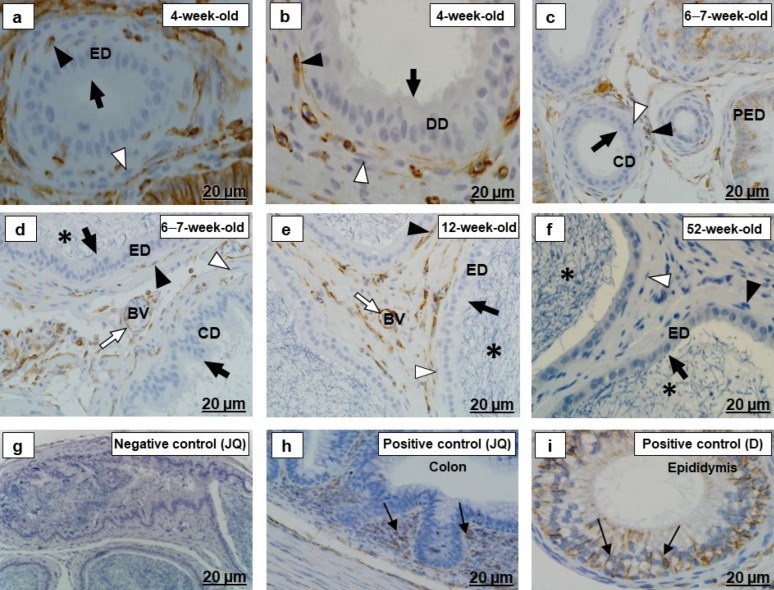
Immunolocalization of vimentin in the epididymal duct unit- (CD) connecting duct, (ED) epididymal duct and (DD) ductus deferens of the testicular excurrent duct system of Japanese quails (*n* = 7 per age group). (**a**–**f**) Immunonegative epithelial- (black arrows) and smooth muscle cells (white arrowheads) of the CD, ED, and DD. (**a**,**b**) Black arrowheads indicate strong immunostaining in the fibroblasts. (**d**,**e**) Moderate immunostaining in the endothelial cells (white arrows) and fibroblasts (black arrowheads). (**f**) Immunonegative fibroblast (black arrowhead). (**d**–**f**) Immunonegative luminal spermatozoa (asterisks). (**g**) Negative control: Efferent ducts and epididymal duct of an adult Japanese quail. (**h**,**i**) Positive controls: Thin black arrows show immunopositive labeling in the lymphocytes in the submucosal layer of the colon of an adult Japanese quail (**h**) and epithelial cells of the dog epididymis (**i**). PED, Proximal efferent duct. BV, Blood vessel. D, Dog. JQ, Japanese quail.

**Figure 11 ijms-23-14028-f011:**
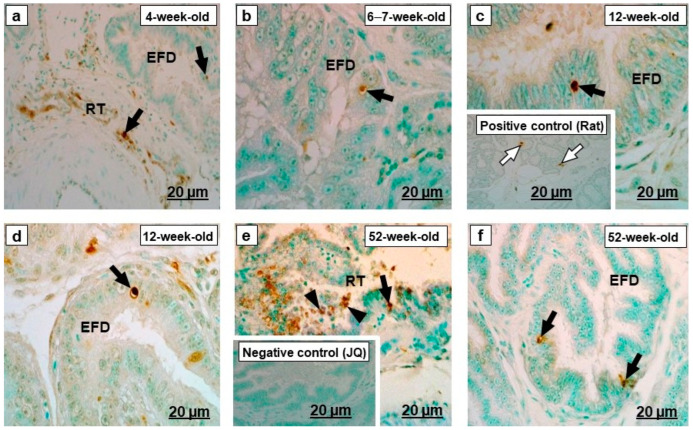
TUNEL-stained sections of the (RT) rete testis and (EFD) efferent ducts of the testicular excurrent duct of Japanese quails (*n* = 3 per age group). (**a**–**f**) Black arrows indicate TUNEL-positive nuclei of epithelial cells in the RT and EFD. Inset (**c**) Positive control: TUNEL-positive cell from the rat mammary gland tissue (white arrowhead). (**e**) Black arrowheads show cellular debris in the RT lumen. (**e**) Inset: Negative control. JQ, Japanese quail.

**Figure 12 ijms-23-14028-f012:**
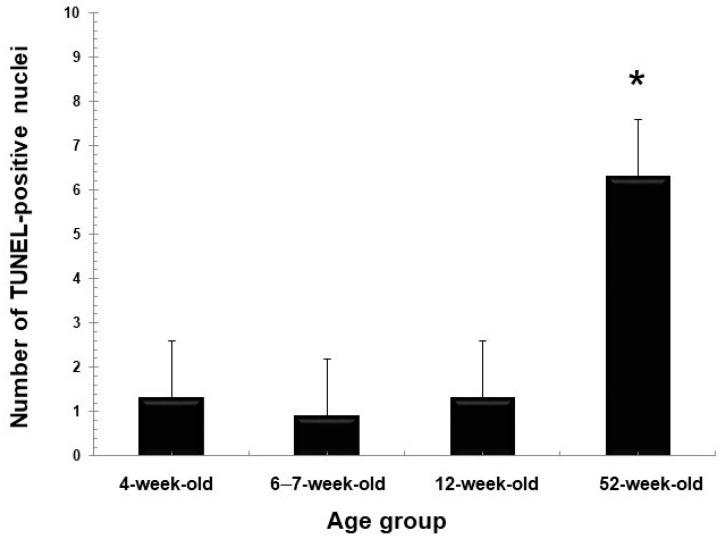
Number of TUNEL-positive nuclei in the epithelia of rete testis and efferent ducts of the testicular excurrent duct system of Japanese quails at 4, 6–7, 12, and 52 weeks of age (*n* = 3 per age group), expressed as mean ± SE. * TUNEL-positive cell counts were significantly (*p* < 0.001) higher in 52-week-old birds compared to the other age groups.

**Table 1 ijms-23-14028-t001:** Antibodies used for immunohistochemistry in this study.

Antibody	Clone	Dilution	Antigen Retrieval	Manufactures	Manufacturer’s Recommended Species Reactivity
Cytokeratin 5	Polyclonal rabbit ab53121	1:50	Citrate buffer (pH 6)	Abcam, Cambridge, United Kingdom	Domestic rabbit, human, mouse, rat
Desmin	Monoclonal mouse ab6322	1:300	Citrate buffer (pH 6)	Abcam, Cambridge, United Kingdom	Human, mouse, rat
Fibronectin	Polyclonal rabbit ab2413	1:250	Tris-EDTA (pH 9)	Abcam, Cambridge, United Kingdom	Human, mouse
Alpha-smooth muscle actin	Monoclonal mouse 1A4	1:50	Citrate buffer (pH 6)	DakoCytomation, Glostrup, Denmark	Avian, chicken, goat, human, mouse, non-human primate, pig, rat, sheep, xenopus
Beta-tubulin	Monoclonal mouse ab44928	1:500	Tris-EDTA (pH 9)	Abcam, Cambridge, United Kingdom	Drosophila, human, melanogaster, mouse, rat,
Vimentin	Monoclonal mouse 3B4	1:25	Citrate buffer (pH 6)	DakoCytomation, Glostrup, Denmark	amphibian, bovine, canine, chicken, human, monkey

**Table 2 ijms-23-14028-t002:** Summary of intensities and immunolocalization of cytokeratin 5 (CK5), alpha-smooth muscle actin (SMA), beta-tubulin (T), and vimentin (V) in the epithelial cells of the testicular excurrent duct system of healthy, captive-bred and raised, male Japanese quails at different age groups (4, 6–7, 12, and 52 weeks of age), *n* = 7 per age group.

Age Group	Epithelium			
	Rete Testis	Proximal Efferent Duct	Distal Efferent Duct	Epididymal Duct Unit
4-week-old	CK5(+++); SMA(+); T(+++); V(+++)	CK5(+++) ^a^; SMA(+); T([+++] ^b^ and [++/+++] ^c^); V(+++)	CK5(+++) ^a^; SMA(+); T([+++] ^b^ and [++/+++] ^c^); V(+++)	CK5(+++); SMA(+); T(+++); V(−)
6–7-week-old	CK5(+++); T(++/+++); V(++/+++)	CK5(++/+++) ^a^; T([+++] ^b^ and [++/+++] ^c^); V(++) ^d^	CK5(++/+++) ^a^; T([+++] ^b^ and [++/+++] ^c^); V(++) ^d^	CK5(+++); T(++); V(−)
12-week-old	CK5(+++); T(++/+++); V(++/+++)	CK5(++/+++) ^a^; T([+++] ^b^ and [++/+++] ^c^); V(++) ^d^	CK5(++/+++) ^a^; T([+++] ^b^ and [++/+++] ^c^); V(++) ^d^	CK5(+++); T(++); V(−)
52-week-old	CK5(+/++); T(+/++); V(+/++)	CK5(+/++) ^a^; T([+++] ^b^ and [+] ^c^); V(+) ^e^	CK5(+/++) ^a^; T([+++] ^b^ and [+] ^c^); V(−)	CK5(++); T(+); V(−)

Intensities of immunoreaction: (−) negative; (+) weak; (++) moderate; (+++) strong. Superscript letters represent the immunostaining—(^a^) apical and subapical regions, (^b^) cilia, (^c^) lateral and subapical regions (^d^) ciliated cells (^e^) few cells.

**Table 3 ijms-23-14028-t003:** Summary of intensities and immunolocalization of the cytokeratin 5 (CK5), desmin (D), fibronectin (F), alpha-smooth muscle actin (SMA), beta-tubulin (T), and vimentin (V) in the peri- and interductal tissue of the testicular excurrent duct system of healthy, captive-bred and raised male Japanese quails at 4, 6–7, 12, and 52 weeks of age (*n* = 7 per age group).

Age Group	Peri- and Interductal Tissues			
	Smooth Muscle Layer	Blood Vessel		Fibroblast
		Endothelial Cell	Tunica media	
4-week-old	CK5(+); D(+++); F(++/+++); SMA(+++); T(++/+++)	CK5(++); F(+++); T(+++); V(+++)	CK5(+); D(++); SMA(+++); T(++)	CK5(−); F(++/+++); T(++/+++); V(+++)
6–7-week-old	CK5(−/+); D(++/+++); F(+/++); SMA(+++); T(++)	CK5(++); F(+++); T(++); V(++)	CK5(+); D(+); SMA(++); T(+)	CK5(−); F(+/++); T(++); V(++)
12-week-old	CK5(−/+); D(++/+++); F(+/++); SMA(+++); T(++)	CK5(++); F(+++); T(++); V(++)	CK5(−/+); D(+); SMA(++); T(+)	CK5(−); F(+/++); T(++); V(++)
52-week-old	CK5(−); D(+/++); T(+/++); F(+); SMA(+++)	CK5(+); F(++); T(+); V(+)	CK5(−); D(+); SMA(++); T(+)	CK5(−); F(+); T(−/+), V(−)

Intensities of immunoreaction: (−) negative; (+) weak; (++) moderate; (+++) strong.

## Data Availability

The authors confirm that the data supporting the findings of this study are available within the article.

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
