# Peer review of "Immunohistochemical Changes in the Testicular Excurrent Duct System of Healthy, Male Japanese Quail (Coturnix coturnix japonica) Observed at 4, 6–7, 12, and 52 Weeks of Age"

_ijms, 2022, doi:10.3390/ijms232214028_

Round 1
Reviewer 1 Report (Previous Reviewer 1)
Authors need to check the manuscript and do a lot of corrections
1. Title: should start with Immunohistochemical.....\
2. Figures 1 and figure 2 & fig 9 and 10 look exactly similar...explain if by mistake correct?
3. Line 87-88: 12 week-old should be adult mice and 52 week-old (early senescence) Japanese quails...correct or justify?
4. Line 407: Mayank et al [36] reported.....Its should be Choubey et al [36] reporter....correct the reference too...all authors first name is written in place of last name..so correct?
Author Response
Reviewer 1
# Comments and Suggestions for Authors
Authors need to check the manuscript and do a lot of corrections
Thank you for the comment. The manuscript has been thoroughly checked.
- Title: should start with Immunohistochemical.....\
The title has been reworded to read: Immunohistochemical changes in the testicular excurrent duct system of healthy, male Japanese quail (Coturnix coturnix japonica) observed at 4, 6-7, 12 and 52 weeks of age
- Figures 1 and figure 2 & fig 9 and 10 look exactly similar...explain if by mistake correct?
Thank you for the comment, the figure has been corrected.
- Line 87-88: 12 week-old should be adult mice and 52 week-old (early senescence) Japanese quails...correct or justify?
The 12 week-old is an adult, this has been corrected by adding (adult) after it.
- Line 407: Mayank et al [36] reported.....Its should be Choubey et al [36] reporter....correct the reference too...all authors first name is written in place of last name..so correct?
The reference has been corrected and crosschecked.

Reviewer 2 Report (Previous Reviewer 2)
The MS has not improved much, There are still too many open, un-answered questions remaining, and the conclusions are based on not robust data. No aging related markers have been provided. The authors mention in the title developemntal stages, what is somehow misleading since the reader would expect to see embryonic stages. Here, the authors compare different age groups.
Author Response
Reviewer 2
#Comments and Suggestions for Authors
The MS has not improved much, There are still too many open, un-answered questions remaining, and the conclusions are based on not robust data. No aging related markers have been provided. The authors mention in the title developmental stages, what is somehow misleading since the reader would expect to see embryonic stages. Here, the authors compare different age groups.
We would like to thank the reviewer for his precious time in reviewing our manuscript. According to the comment above, the manuscript has been changed to reflect immunohistochemical changes in different age groups observed at 4, 6-7, 12, and 52 weeks of age, therefore, the discussion and conclusions were based on comparing the results between these different age groups.
Regarding the developmental stages in the tile: We have reformulated the title to be read as: Immunohistochemical changes in the testicular excurrent duct system of healthy, male Japanese quail (Coturnix coturnix japonica) observed at 4, 6-7, 12 and 52 weeks of age. In addition to that, we have made changes in the manuscript by replacing any sentence containing "developmental stage" with "age group". Therefore, the manuscript can be considered as a comparison study between different age groups.
We are convinced that your comments and suggestions will add value to the manuscript.

Reviewer 3 Report (New Reviewer)
This is an elaborate research article investigating the testicular excurrent duct system of healthy Japanese quail at different developmental stages. The immunohistochemical staining of the cytoskeletal proteins cytokeratin 5, desmin, α-SMA, beta-tubulin, vimentin, and fibronectin, is impressive. Data interpretation is done systematically and the article is well-discussed and concluded with a good data presentation and literature survey. However, there are major flaws in the manuscript that needs to be rectified. The major comments that need to be addressed are as follows:
Major comments:
1. Please mention "n" and p-value under each figure legend.
2. Figure 8. Please provide a different figure as the cy3 intensity is too bright and DAPI is not visible. Please provide the phase contrast image as well for this figure.
3. Why author did not the present the epididymal region sections into a head, body, and tail, and then do the comparative study between the Japanese quail and chicken?
3. Please show apoptotic markers staining or Western blot for each developmental group.
4. Since there is an increase in apoptosis at adult age, was there an increase in ROS?
Author Response
Reviewer 3
#Comments and Suggestions for Authors
This is an elaborate research article investigating the testicular excurrent duct system of healthy Japanese quail at different developmental stages. The immunohistochemical staining of the cytoskeletal proteins cytokeratin 5, desmin, α-SMA, beta-tubulin, vimentin, and fibronectin, is impressive. Data interpretation is done systematically and the article is well-discussed and concluded with a good data presentation and literature survey. However, there are major flaws in the manuscript that needs to be rectified. The major comments that need to be addressed are as follows:
Thank you for your appreciation.
Major comments:
- Please mention "n" and p-value under each figure legend.
Thank you for the comment. The “n” has been added to each figure legend, while the p-value has been added to the figure legend, in which statistical analysis was performed.
- Figure 8. Please provide a different figure as the cy3 intensity is too bright and DAPI is not visible. Please provide the phase contrast image as well for this figure.
Thank you for bringing up this point. We would like to bring this to your attention, the immunofluorescent staining was performed as a confirmation of the beta-tubulin in the excurrent duct system, using an adult Japanese quail and chicken testis as a positive control. The image was captured using an Olympus BX-63 fluorescent microscope, and processed with the ImageJ software. No confocal microscope was used to provide the phase contrast images.
- Why author did not the present the epididymal region sections into a head, body, and tail, and then do the comparative study between the Japanese quail and chicken?
Thank you for the comment. This study was designed based on previous investigations, in which they have described the avian excurrent duct system as a distinct mass of ducts comprising the epididymal region and ductus deferens (Maruch et al., 1998). The epididymal region is a labyrinth of ducts consisting of the rete testis, proximal and distal efferent ducts and connecting ducts (Aire & Soley, 2000; El-Saba & Abdrabou, 2013; Deshmukh et al., 2014). In addition, Aire, (2007) has referred to the connecting duct, epididymal duct and ductus deferens as the epididymal duct unit. Therefore, the avian epididymal region is not divided into a head, body and tail, as in mammals.
- Please show apoptotic markers staining or Western blot for each developmental group.
Thank you for the comment. The TUNEL-stained sections of each age group are presented in Figure 11.
- Since there is an increase in apoptosis at adult age, was there an increase in ROS?
Thank you very much for your question and for bringing up this point. It would have been better to measure the oxidative stress markers and compare them with the frequencies of TUNEL-positive cells in different age groups. Unfortunately, we cannot measure oxidative stress markers due to the unavailability of tissue samples to perform the test. However, we have considered this point as a limitation of the study and added this sentence to the discussion "However, it would have been valuable to study oxidative stress markers and correlate them with the presence of TUNEL-positive cells in these regions at different age groups".
References
Aire, T. A. & Soley, J. The surface features of the epithelial lining of the ducts of the epididymis of the ostrich (Struthio camelus). Anatomia, Histologia, Embryologia, 2000, 29, 119-126.
Aire, T. A. Anatomy of the testis and male reproductive tract. In: Jamieson, B. G. M. (ed.) Reproductive Biology and Phylogeny of Aves of Birds. 1st ed. Science Publisher, 2007, Enfield, New Hampshire.
Deshmukh, S., Ingole, S., Chaurasia, D., Karmore, S. & Sinha, B. Comparative histological and histochemical studies on epididymis of Aseel and Vanaraja breeds of poultry. Indian Journal of Veterinary Anatomy, 2014, 26, 103-104.
El-Saba, A. A. & Abdrabou, M. I. Histological and Ultrastructural studies On the Epididymis of Pigeon (Columba livia). Nature and Science, 2013, 11(10), 53-63.
Maruch, S. M. d. G., Ribeiro, M. D. G. & Teles, M. E. D. O. Morphological and histochemical aspects of the epididymal region and ductus deferens of Columbina talpacoti (Temminck)(Columbidae, Columbiformes). Revista Brasileira de Zoologia, 1998 15, 365-373.

Round 2
Reviewer 3 Report (New Reviewer)
The authors have addressed all the comments diligently. However, I just wanted to point out that phase contrast by fluorescent microscope can be done by changing the condenser ring to PH1/PH2. If you don't have that, you can always provide a bright field microscope to avoid any contradiction for the artifacts.
This manuscript is a resubmission of an earlier submission. The following is a list of the peer review reports and author responses from that submission.
Round 1
Reviewer 1 Report
The Ibrahim et al., 2022, Manuscript ID: ijms-1839898 addresses the age-related testicular changes in the excurrent duct system of male Japanese quail. A search on Pubmed.gov for the terms "Testis" and "aging" and "excurrent duct" keywords resulted in only no hits that depicts the novelty of this study.
There are few important queries and few suggestion which makes this manuscript more representable to be publish.
1. Can the authors include the immunoblotting changes (western blot images) to quantify the changes of proteins (cytokeratin 5, desmin, fibronectin, α-SMA and beta-tubulin) during aging in testicular excurrent ducts?
2. Can the authors justify why they have not selected prenatal mice in this study? If it is age-related changes then you have to consider the starting age too? If possible can you check and cite the MS “Role of adiponectin as a modulator of testicular function during aging in mice” and “Adiponectin/AdipoRs signaling as a key player in testicular aging and associated metabolic disorders” consider to incorporate those age groups.
3. Do you have any idea about the changes in the expression of obesity hormone adiponectin in the testicular excurrent ducts? It will be more informative if you correlate cytokeratin 5, desmin, fibronectin, α-SMA and beta-tubulin proteins with adiponectin. It will be very good study or atleast for your future study?
4. Can the authors justify why they didn’t quantified the proteins through there IHC images using softwares like imageJ?
Author Response
Reviewer 1
# Comments and Suggestions for Authors
The Ibrahim et al., 2022, Manuscript ID: ijms-1839898 addresses the age-related testicular changes in the excurrent duct system of male Japanese quail. A search on Pubmed.gov for the terms "Testis" and "aging" and "excurrent duct" keywords resulted in only no hits that depicts the novelty of this study.
Thank you very much for the comment. Yes, we do agree with you, there are several studies regarding the excurrent duct of the testis, however, most of these studies have been conducted on the histology, ultrastructure and immunohistochemistry of various adult avian species, including Japanese quail. Although the Japanese quail is a useful model to study the effects of endocrine-disrupting compounds (EDCs), there are no reports on the morphology and immunohistochemistry of the testicular excurrent duct system during the different reproductive stages including pre-puberty, puberty and aged. Therefore, we think this study makes a valuable contribution and serves as a baseline normal for testicular excurrent duct in healthy birds at different developmental stages, for comparison in future, similar immunohistochemical studies involving environmental toxins affecting this region.
# There are few important queries and few suggestion which makes this manuscript more representable to be publish.
- Can the authors include the immunoblotting changes (western blot images) to quantify the changes of proteins (cytokeratin 5, desmin, fibronectin, α-SMA and beta-tubulin) during aging in testicular excurrent ducts?
Thank you very much for bringing up this point. We do agree with you it would have been better to do quantify the proteins with western bolt tests. Prior to the rest of the study we tried to quantify these proteins, however, the tissue is limited and the size of this region is too small to perform the western blot. To circumvent this problem, we used positive and negative controls for each antibody from normal chicken and domestic canine. In addition, we added this point as a limitation of the study “Other testing modalities such as immunoblots would be required to ratify the presence of these proteins”.
- Can the authors justify why they have not selected prenatal mice in this study? If it is age-related changes then you have to consider the starting age too? If possible can you check and cite the MS “Role of adiponectin as a modulator of testicular function during aging in mice” and “Adiponectin/AdipoRs signaling as a key player in testicular aging and associated metabolic disorders” consider to incorporate those age groups.
Thank you for the comment. We have selected the Japanese quail because it has many advantages as an avian model due to the ease of husbandry, the small size, and the short time to sexual maturity as well as its suitability to study the effects of endocrine disrupting chemicals (EDCs) during sexual differentiation and maturation. This study is part of a project to investigate short and long-term exposures to EDCs and evaluate the effects on the cytoskeletal proteins and the reproductive behaviour of quail. There is a dearth of information on the normal distribution of cytoskeletal proteins during different developmental stages.
However, we have included the reference to adiponectin’s role in aging. Mayank et al [32] reported a decrease in adiponectin, and its receptors expression and testosterone synthesis in the testis of aged mice, which leads to an increase oxidative stress and alters testicular functions.
Do you have any idea about the changes in the expression of obesity hormone adiponectin in the testicular excurrent ducts? It will be more informative if you correlate cytokeratin 5, desmin, fibronectin, α-SMA and beta-tubulin proteins with adiponectin. It will be very good study or atleast for your future study?
Thank you for the question. It will be very interesting to know the normal distribution and the role of adiponectin in the excurrent duct system, because this region is playing an important role in the transportation, maturation and viability of spermatozoa. However, to the best of our knowledge there is no publication regarding the distributions and immunolabelling in this region. We will certainly attempt to include this in our future studies, thanks again for the suggestion.
- Can the authors justify why they didn’t quantified the proteins through there IHC images using softwares like imageJ?
Thank you very much for raising this point. In this study, we have used visual examination rather than ImageJ software because according to the ImageJ manual, we need to split multi-colour images into single channels and convert single channel colour images to grayscale before proceeding. We have used an immunohistochemistry image, which cannot be split like the immunofluorescence.

Reviewer 2 Report
The manuscript has been evaluated carefully. Although the topic appears to be interesting, this reviewer has critical concerns with regards to the study design. First of all, seven individuals are a bit low "n" in each age group, what could be also the reason why there was a variation in intensities of immunolabelling and distribution of cytoskeletal proteins and fibronectin in epithelial cells and tissues of the testicular excurrent duct system. Secondly, to really investigate an evident aging effect the "aged group" should be older at least 18months or close to 24 months. As a third point, it would be interesting to know if known markers for male reproductive aging (for instance RIPK ort SIRT) decrease with age, to really confirm that aging takes place in the collected tissues.
Author Response
Reviewer 2
#Comments and Suggestions for Authors
The manuscript has been evaluated carefully. Although the topic appears to be interesting, this reviewer has critical concerns with regards to the study design. First of all, seven individuals are a bit low "n" in each age group, what could be also the reason why there was a variation in intensities of immunolabelling and distribution of cytoskeletal proteins and fibronectin in epithelial cells and tissues of the testicular excurrent duct system. Secondly, to really investigate an evident aging effect the "aged group" should be older at least 18months or close to 24 months. As a third point, it would be interesting to know if known markers for male reproductive aging (for instance RIPK ort SIRT) decrease with age, to really confirm that aging takes place in the collected tissues.
The authors would like to thank the reviewer for the valuable time to evaluating this work. G*power statistical program has been used to determine the sample size of the quantitative data (number of apoptotic nuclei in the epithelium of the rete testis and efferent ductules of the Japanese quail's age group). The program showed a 90 - 95% chance of correctly rejecting the null hypothesis of no difference with a total of 24 – 28 birds. Therefore, the sample size (28 birds) in the present study has been selected according to maximum power (0.95 power at P a = 0.05). In addition, the Animal Ethics Committee, based on the estimation of the sample size and in an endeavour to reduce the number of sentient animals being used, only approved the use of 7 birds per group.
F tests - ANOVA: Fixed effects, omnibus, one-way
Analysis: A priori: Compute required sample size
Input: Effect size f = 0.8516505
α err prob = 0.05
Power (1-β err prob) = 0.90
Number of groups = 4
Output: Noncentrality parameter λ = 17.4074058
Critical F = 3.0983912
Numerator df = 3
Denominator df = 20
Total sample size = 24
Actual power = 0.9020925
F tests - ANOVA: Fixed effects, omnibus, one-way
Analysis: A priori: Compute required sample size
Input: Effect size f = 0.8516505
α err prob = 0.05
Power (1-β err prob) = 0.95
Number of groups = 4
Output: Noncentrality parameter λ = 20.3086401
Critical F = 3.0087866
Numerator df = 3
Denominator df = 24
Total sample size = 28
Actual power = 0.9502981
The sample size (28 birds) of the immunohistochemistry has been determined according to previous publications in such topics (Tamilselvan et al 2021, Zakariah et al 2020, Abdul-Rahman and Jeffcoate, 2018). Therefore, a total of 28 Japanese quail birds (7 birds/group) has been taken for this study.
- Regarding the second point “to really investigate an evident aging effect the "aged group" should be older at least 18months or close to 24 months”?
Thank you very much for the comment. The aging birds have been selected based on our previous work "Apoptosis of germ cells in the normal testis of the Japanese quail (Coturnix Coturnix japonica)”, in which changes were observed in the aged bird (52 weeks old).
- As a third point, it would be interesting to know if known markers for male reproductive aging (for instance RIPK ort SIRT) decrease with age, to really confirm that aging takes place in the collected tissues.
Thank you for the comment. It would have been interesting to investigate the markers for male reproductive aging. However, in this study, we have focused on the age-related changes in the cytoskeletal proteins and fibronectin in the excurrent duct system of quail to provide baseline information for future studies on the effect of the endocrine disrupters on the male reproductive behaviour using quail as an animal model.
Reference
Zakariah, M, Ibrahim MI, Molele RA and McGaw LJ Apoptosis of germ cells in the normal testis of the Japanese quail (Coturnix coturnix japonica), Tissue and Cell 2020 67:101450.
Abdul-Rahman, II and Jeffcoate I Histological structure and age-related changes in the luminal diameter of the excurrent duct system of guinea cocks (Numida meleagris) and associated changes in testosterone concentrations, Canadian Journal of Veterinary Research 2018 82:60-65.
Tamilselvan, S, Dhote B and Sinha R Age wise changes in morphology and morphometry of epididymis of Guinea fowl (numida meleagris), Journal of Experimental Zoology 2021 24:79-82.

Round 2
Reviewer 1 Report
The authors have justified the queries raised and try to put the limitations of the study. Also they have included the positive and negative controls for IHC images.
Comments: Can the authors cite the the article relevant to in the discussion section “Adiponectin/AdipoRs signaling as a key player in testicular aging and associated metabolic disorders”
Reviewer 2 Report
As a specialist in reproductive aging, I am not convinced that 52 weeks old birds are "aged" enough to show real aging related changes. In another recent paper ( please see below), the aged group was set at 144 weeks. Therefore, the whole study cannot be evaluated with regards to aging!
Age dependent variations in the deep brain photoreceptors (DBPs), GnRH-GnIH system and testicular steroidogenesis in Japanese quail, Coturnix coturnix japonica
-
PMID: 29580815
-
DOI: 10.1016/j.exger.2018.03.018